# Effect of Sintering Temperature on Adhesion Property and Electrochemical Activity of Pt/YSZ Electrode

**DOI:** 10.3390/ma15103471

**Published:** 2022-05-12

**Authors:** Jixin Wang, Jiandong Cui, Xiao Zhang, Wentao Tang, Changhui Mao

**Affiliations:** 1GRINMAT State Key Laboratory of Advanced Materials for Smart Sensing, GRINM Group Co., Ltd., Beijing 100088, China; wangjixin68@163.com (J.W.); cuijiandong@grinm.com (J.C.); zhangxiao@grinm.com (X.Z.); youyantangwentao@126.com (W.T.); 2GRIMAT Engineering Institute Co., Ltd., Beijing 101407, China; 3General Research Institute for Nonferrous Metals, Beijing 100088, China

**Keywords:** NO_x_ sensor, Pt/YSZ electrode, sintering temperature, electrochemical activity, electrochemical analysis

## Abstract

The (Pt/YSZ)/YSZ sensor unit is the basic component of the NO_x_ sensor, which can detect the emission of nitrogen oxides in exhaust fumes and optimize the fuel combustion process. In this work, the effect of sintering temperature on adhesion property and electrochemical activity of Pt/YSZ electrode was investigated. Pt/YSZ electrodes were prepared at different sintering temperatures. The microstructure of the Pt/YSZ electrodes, as well as the interface between Pt/YSZ electrode and YSZ electrolyte, were observed by SEM. Chronoamperometry, linear scan voltammetry, and AC impedance were tested by the electrochemical workstation. The results show that increasing the sintering temperature (≤1500 °C) helped to improve adhesion property and electrochemical activity of the Pt/YSZ electrode, which benefited from the formation of the porous structure of the Pt/YSZ electrode. For the (Pt/YSZ) electrode/YSZ electrolyte system, O^2−^ in YSZ is converted into chemisorbed O_2_ on Pt/YSZ, which is desorbed into the gas phase in the form of molecular oxygen; this process could be the rate-controlling step of the anodic reaction. Increasing the sintering temperature (≤1500 °C) could reduce the reaction activation energy of the Pt/YSZ electrode. The activation energy reaches the minimum value (1.02 eV) when the sintering temperature is 1500 °C.

## 1. Introduction

Emission gases, including oxides of carbon, oxides of nitrogen and oxides of sulfur, have been a highly regarded research area because of the growing awareness of environmental protection. Nitrogen oxides (mainly NO and NO_2_) bring acid rain and photochemical smog, which pose a great threat to human health and environmental safety [1]. NO_x_ sensor is a key device to control this problem by monitoring the NO_x_ content in exhaust gas and optimizing the fueling combustion process [2,3,4]. At present, NO_x_ sensors are mainly divided into the following four types: potential type, mixed potential type, complex impedance type and current type [5,6,7,8], of which the current type sensor is the only one commercially used until now. The structure of this type of NO_x_ sensor is shown in Figure 1a which contains a small hole in the left side for the exhaust gas to enter. This sensor mainly consists of two cavities and three oxygen-pumping cells. The pump oxygen battery has a (Pt/YSZ) electrode/YSZ electrolyte sensor unit structure, which consists mainly of an oxygen-vacancy-rich YSZ electrolyte and two highly electrochemically active Pt/YSZ electrode. The detection process is shown in Figure 1b,c. The main pump and auxiliary pump fully pump the O_2_ in the exhaust gas in the two cavities, inducing the conversion of NO_2_ into NO, and NO is finally decomposed into N_2_ and O_2_. When the decomposed O_2_ is pumped away by the measuring pump oxygen cell, the concentration of decomposed O_2_ can be obtained by measuring the corresponding pump current, and the NO_x_ concentration can be obtained after conversion. In this NO_x_ sensor unit structure, the interface matching of YSZ solid electrolyte and Pt/YSZ electrode, as well as the electrochemical activity of Pt/YSZ electrode, seriously affect the performance of NO_x_ sensors. Moreover, the electrode composition, electrode thickness, sintering process, microstructure, and morphology are all key factors for the Pt/YSZ electrode [9,10,11,12]. Jaccoud et al. [10] found that the Pt electrode prepared by Pt slurry had better electrochemical performance than that prepared by sputtering. Nurhamizah [13] found that the Pt electrode with porous structure has better electrochemical performance. Boer [14] and Xia [15] found that a certain proportion of YSZ powder in the Pt slurry could improve the porous structure of the Pt electrode and improve the adhesion of the Pt electrode to the YSZ electrolyte. Li et al. [16] found that properly increasing the sintering temperature could promote the electrode to obtain greater electrochemical activity and more three-dimensional network electrode structure. In most of the previous research, YSZ green tapes were firstly sintered to high density ceramic as electrolyte, on which the Pt/YSZ electrode slurry was printed and finally sintered together. However, this method is not suitable for NO_x_ sensors with multilayer ceramic structures and different functional electrodes. For the (Pt/YSZ)/YSZ sensor unit, Pt/YSZ electrode slurry is firstly printed on the surface of YSZ green tapes, and then followed by high temperature co-sintering. In general, defects such as warpage, cracking and delamination would be the main challenge in the sintering process. The problem of inconsistent sintering shrinkage between electrode and electrolyte is a major difficulty in the research of Pt/YSZ electrode. Another difficulty is ensuring the excellent electrochemical activity of the Pt/YSZ electrode. It is well known that increasing sintering temperature, as a key part of the electrode preparation, has huge impact on electrode microstructure and electrochemical performance [17]. The properties of Pt/YSZ electrodes printed on sintered YSZ ceramics are widely studied, but there are few systematic reports on the effect of sintering temperatures above 1400 °C on adhesion property and electrochemical activity of Pt/YSZ electrodes [12,18,19]. In this experiment, the effect of sintering temperature up to 1550 °C on the performance of Pt/YSZ electrode was studied. The chrono-current, linear scanning voltammetry and AC impedance of the Pt/YSZ electrodes were tested by an electrochemical workstation. The microstructure of the Pt/YSZ electrodes, as well as the interface between Pt/YSZ electrode and YSZ electrolyte, were also observed by SEM.

## 2. Experimental Procedure

The YSZ green tapes were prepared by tape casting process. Triethanolamine (AR), polyvinyl butyral (PVB, AR), polyethylene glycol (PEG, AR) and dibutyl phthalate (DBP, AR) were used as dispersant, binder and plasticizer, respectively. Ethanol (AR) and butanone (AR) were used as solvent. The above reagents were purchased from Sinopharm Chemical Reagent Co., Ltd. The physical properties of YSZ and Pt materials were summarized in Table 1. The Pt/YSZ electrode slurry (Pt + 15 wt.% YSZ powder) was provided by GRINMAT Engineering Institute Co., Ltd. The Pt/YSZ electrode slurry was printed on the YSZ green tapes, and the (Pt/YSZ)/YSZ sensor units were obtained by sintering at different sintering temperatures under air condition. The schematic diagram of the sample preparation process of the (Pt/YSZ)/YSZ sensor unit is shown in Figure 2. The microscopic morphology of the samples was observed by SEM (JSM-7610F, Japan). The electrical properties of the Pt/YSZ electrodes were tested by a CHI660D electrochemical workstation, the mixture of gases with 10 vol.% O_2_ and 90 vol.% N_2_ was added during the test. Specific test parameters were as follows: (1) Chrono-amperometric experiment: set a fixed voltage of 600 mV between the two poles, the scanning time was 0–180 s, the test temperature was 750 °C, (2) Linear scan voltammetry test: the scanning voltage was −0.6 V to 0.6 V, the test temperature was 750 °C, (3) AC Impedance (EIS) Test: Set the frequency range to 0.001 Hz–10 MHz, the signal voltage was 500 mV, the test temperatures were 600 °C, 650 °C, 700 °C, 750 °C and 800 °C, respectively.

## 3. Results and Discussions

### 3.1. Micromorphologies of Pt/YSZ Electrodes

The micromorphology of the Pt/YSZ electrodes under different sintering temperatures is shown in Figure 3. The YSZ grains are dispersed among the Pt grains, which reduces the aggregation of Pt grains and promotes the formation of a porous structure in Pt/YSZ electrode [20]. When the sintering temperature reaches 1550 °C, the Pt grains gradually grow, aggregate and even melt together, which destroys the porous structure of the Pt/YSZ electrode. It was reported that extremely high temperature could cause over-sintering of the electrolyte material, which results in the pore size and porosity decrease, and the active area of electrode reaction presents a first increasing and then decreasing trend [12,16]. When the sintering temperature decreases to 1500 °C, the large particles of Pt and the small particles of YSZ are uniformly mixed together to form an optimal porous electrode structure. With the comparative analysis, it can be concluded that the Pt/YSZ electrode with a sintering temperature of 1500 °C has a better porous structure, which would benefit gas transmission and NO_x_ evaluation process.

Figure 4 shows the EDS analysis of Pt/YSZ electrodes prepared at different sintering temperatures. As the characteristic X-ray peaks of Zr and Pt are similar (Zr (Lα2.0424) and Pt (Mα2.0485)), it is difficult to distinguish these two elements by EDS. Zr and Y are solid-dissolved together. In this paper, Y element is used instead of YSZ to interpret the composition of the YSZ and Pt/YSZ.

### 3.2. Adhesion Analysis of Pt/YSZ Electrodes

It is necessary to evaluate the electrode adhesion behavior because the interface mismatch between the Pt/YSZ electrode and the YSZ electrolyte sintered at a high temperature seriously affects the performance of the (Pt/YSZ)/YSZ sensor unit. The Pt/YSZ electrode adhesion was tested using an ultrasonic cleaner (QR-020S, 40 kHz and 120 W). The Pt/YSZ electrode was then placed in a cleaner filled with distilled water and the weight of each sample was measured before and after the test over a test period of 10 min to 90 min. The samples were dried at 120 °C for 30 min before the sample weight was measured. The weight loss of the electrode is calculated as follows [21]:(1)$$\Delta W=\left[\frac{\left(W_{1}-W_{2}\right)}{\left(W_{1}-W_{0}\right)}\right]\times 100\%$$

Among them, $W_{0}$, $W_{1}$ and $W_{2}$ represent the weight of the YSZ electrolyte and the weight of the Pt/YSZ electrode before and after ultrasonic vibration, respectively.

Figure 5 shows the weight loss of Pt/YSZ electrodes prepared at different sintering temperatures. With the increase of sintering temperature, the weight loss of Pt/YSZ electrodes reaches smaller, which means that increasing the sintering temperature significantly improves the adhesion of Pt/YSZ electrodes.

The matching results under different sintering temperatures can also be reflected in the micro-morphology of the interface between the Pt/YSZ electrode and the YSZ electrolyte, as shown in Figure 6. Combining the secondary electron image and the backscattered electron image, the interpenetrating structure between YSZ particles and Pt particles with a relatively complete interface are clearly observed. At lower sintering temperatures, the flat interface between the electrode and the electrolyte can be clearly observed. It can be explained that the lower sintering temperature cannot provide sufficient diffusion driving force to promote the interdiffusion of YSZ particles and Pt particles. With the increase of the sintering temperature, the grains of YSZ and Pt gradually grow up, the grain boundaries migrate, and the degree of bonding between the grains is significantly enhanced. Combined with the morphologies of Pt/YSZ electrodes in Figure 3, it can be found that the increase of the sintering temperature is beneficial to the sintering matching of the electrode and the solid electrolyte. However, when the temperature reaches 1550 °C, the over-sintering phenomenon of the Pt particles and YSZ particles is very obvious.

### 3.3. Chronoamperometry

A constant potential is applied to the Pt/YSZ electrodes to obtain a current-time curve, which is called chrono-amperometry. The stability of the electrode can be investigated by observing the change of the current value, and the electrochemical activity can be analyzed by comparing the stable current value [22,23]. The chrono-amperometric curves at different sintering temperatures are shown in Figure 7. Each curve shows the same trend of change, and the current reaches a stable value in a relatively short period of time. In NO_x_ testing process, the anodic reaction model of the Pt/YSZ electrode system can be expressed as [24]:(2)$$O^{2-}\overset{-2e^{-}}{\to }O_{\mathrm{atoms}}\overset{+{\mathrm{Pt}}_{\mathrm{atoms}}}{\to }\mathrm{Pt}­O$$
(3)$$O^{2-}\overset{-2e^{-}}{\to }O_{\mathrm{atoms}}\to \frac{1}{2}O_{2\left(\mathrm{gas}\right)}$$

When the electrode was anodically polarized, O^2−^ in YSZ electrolyte diffused to the surface of Pt/YSZ electrode, releasing two electrons to generate oxygen atoms. A portion of the oxygen reached the Pt/YSZ electrode and reacted with the Pt_atoms_ to form PtO. The electrical conductivity of PtO is very poor, which hinders the charge transfer process, thus the current density decreases rapidly in a short time [25]. The number of platinum atoms on the electrochemical reaction site is limited, and the reaction reaches saturation in a short time [24]. The other portion O^2−^ is carried out by reaction (3), that is, the reaction moves to the interface between the Pt/YSZ electrode and the YSZ electrolyte, and O^2−^ in YSZ is converted into chemisorbed O_2_ on Pt/YSZ. It is desorbed into the gas phase in the form of molecular oxygen, which does not accumulate at the Pt/YSZ electrode, and the reaction can proceed infinitely, as well as obtain a stable current density. The tendency of the current to decrease first and then stabilize may be the result of the combined effect of the two reaction models. The possible anodic reaction model of the Pt/YSZ electrode is shown in Appendix A. As the sintering temperature gradually increases, the current of the Pt/YSZ electrode increases first and reaches the maximum value at 1500 °C, followed by sharp current decrease at 1550 °C.

### 3.4. Linear Scan Voltammetry Analysis

In a certain potential range, apply a continuous triangular wave signal to the Pt/YSZ electrodes, scan from the cathode direction to the anode direction with a constant scanning rate, and record the curve of current versus voltage, which is called linear scan voltammetry (LSV) [26]. By analyzing the cathodic and anodic peaks of the linear voltammetry curve, the possible electrode reactions, the reversibility of the electrode reactions and the source of the reaction products can be studied. The rate of the electrode reactions can be evaluated by comparing the slopes of the curves. The LSV curves of the Pt/YSZ electrodes are shown in Figure 7. No obvious cathodic peaks could be observed, but anodic peaks could be observed at about 0.5 V, which is consistent with previous research [27]. Appendix A shows the LSV curve of the Pt/YSZ electrode when the scan voltage is expanded to −2 V, but there was still no current saturation plateau. In the Pt/YSZ electrode reaction system, the anodic peak appears because the anodic reaction (1) occurs, where the poor conductive Pt-O produced by the reaction accumulates on the Pt/YSZ electrode, hindering the charge transfer process, which also explains the phenomenon of the current drop shown in Figure 7. No cathodic peak was observed, one possible explanation is that this process of releasing oxygen does not seem to be related to any electrochemical process, but a chemical reaction [28]:(4)$${\mathrm{PtO}}_{x}\to \frac{x}{2}O_{2\left(g\right)}+\mathrm{Pt}$$

Appendix A shows the photoelectron spectrum obtained from the surface of the Pt/YSZ electrode, only Pt peaks were observed, and no PtO peaks were observed. Although no obvious cathodic peak is observed, the current changed drastically with increasing potential, approximately fitting the curve of the cathodic reaction to a straight line, and the fitting results are shown in Figure 8. It can be seen that these slopes gradually increase as the sintering temperature increases. The slope relationship of these curves is: 1500 °C > 1450 °C > 1550 °C > 1400 °C > 1350 °C. It can be inferred that the cathodic reaction rate of the Pt/YSZ electrode is the fastest when the sintering temperature is 1500 °C, indicating the highest electrochemical catalytic activity.

### 3.5. AC Impedance Spectrum Analysis

The AC impedance spectrums of the Pt/YSZ electrodes sintered at different temperatures tested at 750 °C is shown in Figure 9. The equivalent circuit is shown in Figure 10, which includes two RCPE elements (R_se_ and R_ct_) connected in series with R_0_. R_0_ represents the wire resistance, which is not displayed in the AC impedance spectrum. The R_se_ element corresponds to the first small semicircle in the AC impedance spectrum and is located in the high frequency region, usually representing the resistance of the YSZ electrolyte. R_ct_ corresponds to the second semicircle in the AC impedance spectrum, which is located in the low frequency region and represents the Pt/YSZ electrode resistance. The spectrum can be used to express the difficulty of charge transfer across the interface between the electrode and the electrolyte solution during the electrode reaction process [29]. Through the AC impedance, the resistances of the electrolyte and electrode and the activation energy of the reaction can be calculated respectively. The smaller the resistance is, the easier the charge transfer process will become. And the smaller the reaction activation energy is, the higher the electrochemical activity of the electrode will be [30,31]. The diameter of the semicircle on the AC impedance spectrum represents the size of the resistance, and a small diameter is recommended. Obviously, increasing sintering temperature (≤1500 °C) reduces the resistance of the electrodes. However, the electron migration becomes difficult while the sintering temperature reaches 1550 °C. A possible explanation for this phenomenon is that the destroyed porous structure for Pt/YSZ electrode, resulting an abnormal increase in resistance.

According to the Arrhenius equation, the variation of the conductivity of the sample with the test temperature (600 °C, 650 °C, 700 °C, 750 °C and 800 °C, respectively) can be expressed as [32]:(5)$$\sigma =\frac{A}{T}e^{-\frac{E}{kT}}$$
transform it into another form:(6)$$\ln\sigma T=-\frac{E}{kT}+\ln A$$

Among them: σ is the conductivity (mS·cm^−1^), *T* is the test temperature (K), *k* is Boltzmann constant (1.38 × 10^−23^ J/K), *E* is the activation energy (eV), *A* is a constant. Therefore, taking 1000/*T* as the abscissa and lnσT as the ordinate to draw a curve, according to the fitting slope of the obtained curve, the specific value of the diffusion activation energy can be calculated. The Arrhenius curves of Pt/YSZ electrodes at different sintering temperatures are shown in Figure 11. And the activation energy of the electrode reaction can be obtained by linear fitting of each point, as shown in Table 2. With the increase of sintering temperature, the electrode activation energy first increases and then decreases, and the electrode activation energy decreases to the smallest value at 1500 °C (1.02 eV).

## 4. Conclusions

In this study, Pt/YSZ electrodes were prepared by a conventional sintering method, and the effects of sintering temperature on the adhesion properties, micromorphology, and electrochemical activity of Pt/YSZ electrodes were investigated. Increasing the sintering temperature (≤1500 °C) helps to improve the adhesion properties and electrochemical activity of Pt/YSZ electrodes, and is beneficial to the formation of porous structures of Pt/YSZ electrodes. For the (Pt/YSZ) electrode/YSZ electrolyte system, O^2^^−^ in YSZ is converted into O_2_ chemisorbed on Pt/YSZ, desorbed into the gas phase in the form of molecular oxygen, this process may be a rate-controlled anodic reaction control Step. Increasing the sintering temperature (≤1500 °C) reduces the reaction activation energy of the Pt/YSZ electrode. The activation energy reaches a minimum value (1.02 eV) when the sintering temperature is 1500 °C.

## Figures and Tables

**Figure 1 materials-15-03471-f001:**
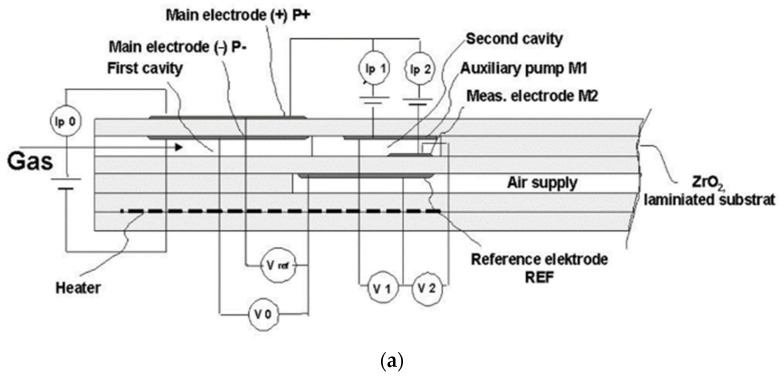
(**a**) Structure of the current type NO_x_ sensor; (**b**) NO_x_ concentration detection principle; (**c**) (Pt/YSZ)/YSZ sensor unit oxygen pumping principle.

**Figure 2 materials-15-03471-f002:**
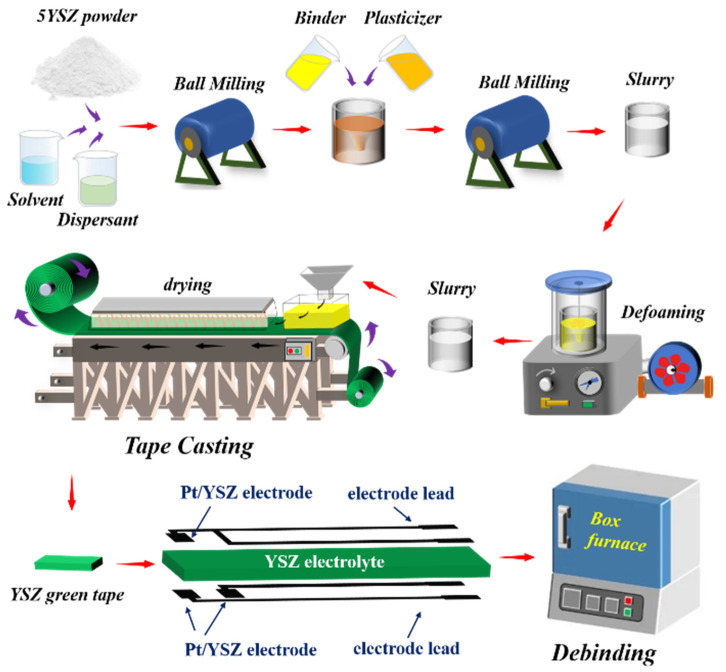
Sample preparation process of the (Pt/YSZ)/YSZ sensor unit.

**Figure 3 materials-15-03471-f003:**
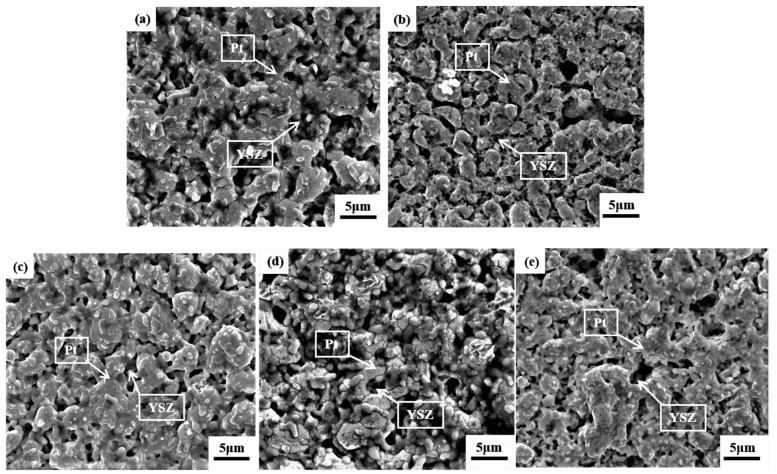
SEM morphologies of Pt/YSZ electrodes at different sintering temperatures (**a**) 1350 °C, (**b**) 1400 °C, (**c**) 1450 °C, (**d**) 1500 °C and (**e**) 1550 °C.

**Figure 4 materials-15-03471-f004:**
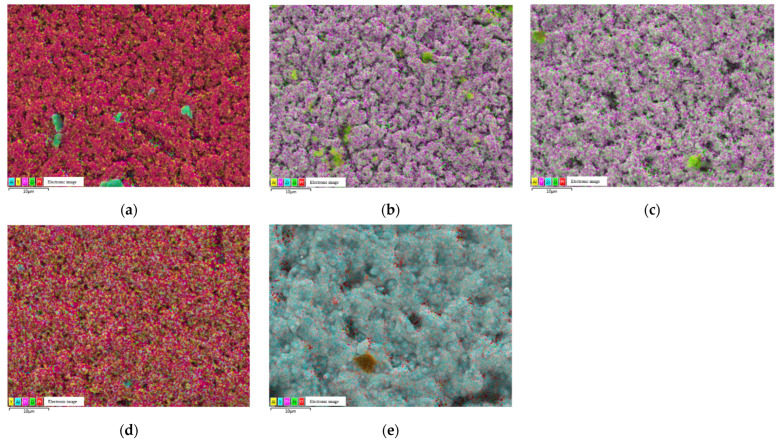
Energy Dispersive X-ray Spectroscopy (EDS) images of Pt/YSZ electrodes fabricated at different sintering temperatures (**a**) 1350 °C, (**b**) 1400 °C, (**c**) 1450 °C, (**d**) 1500 °C and (**e**) 1550 °C.

**Figure 5 materials-15-03471-f005:**
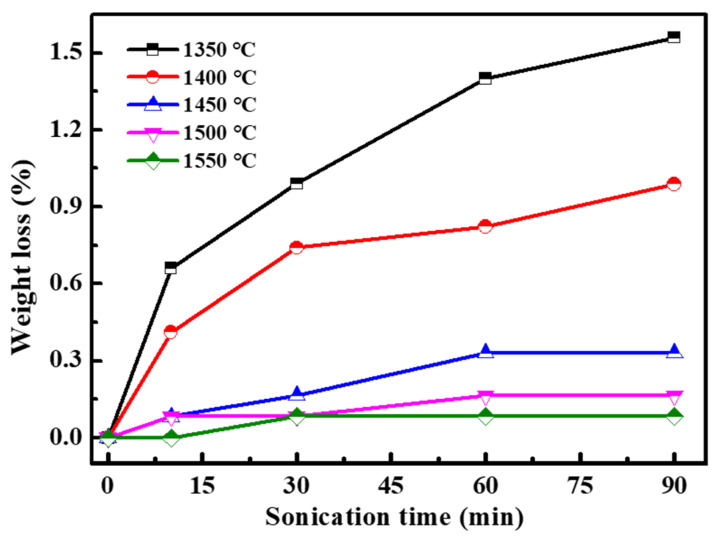
Weight loss of Pt/YSZ electrodes prepared at different sintering temperatures.

**Figure 6 materials-15-03471-f006:**
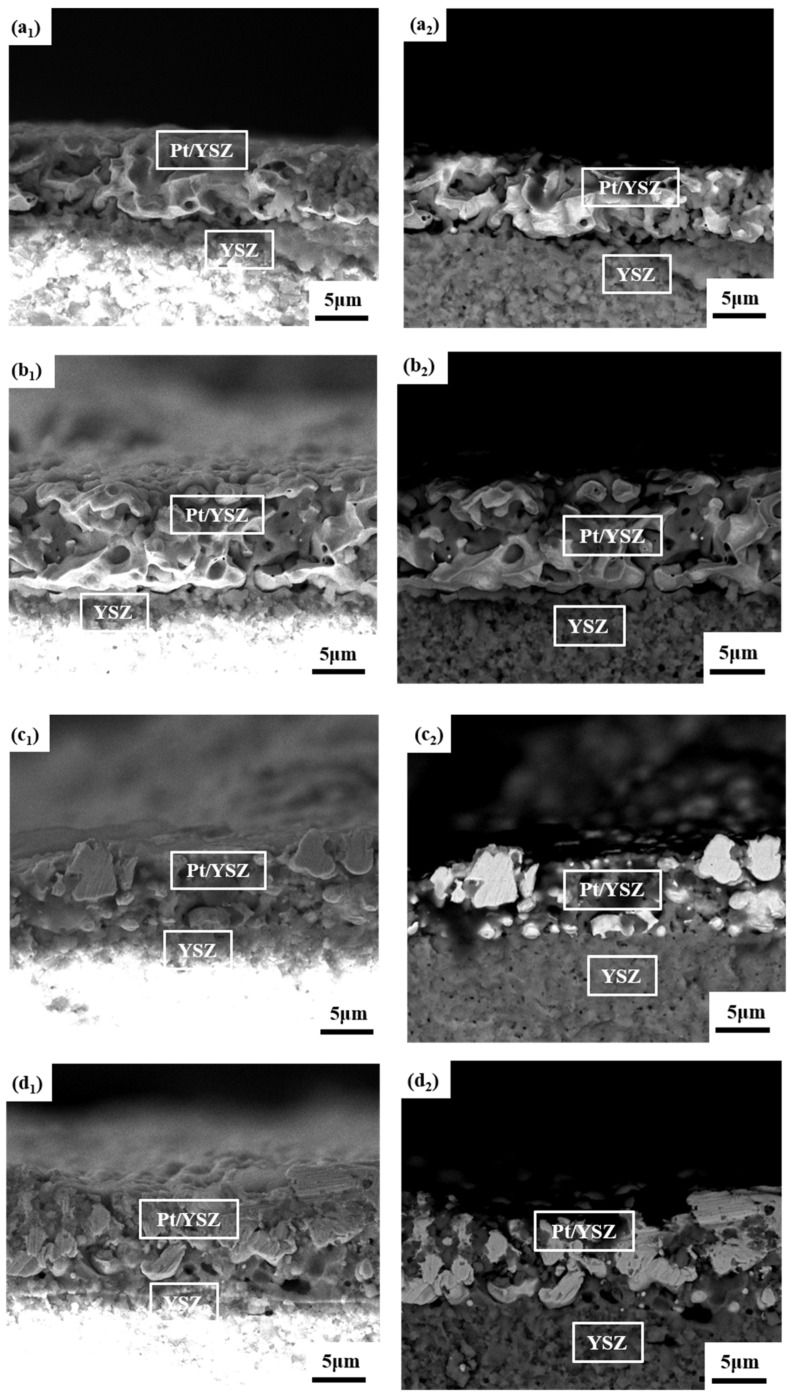
SEM morphologies of the interface of the Pt/YSZ electrodes and the YSZ electrolyte at different sintering temperatures. secondary electron image: (**a_1_**) 1350 °C, (**b_1_**) 1400 °C, (**c_1_**) 1450 °C, (**d_1_**) 1500 °C, (**e_1_**) 1550 °C. backscattered electron image: (**a_2_**) 1350 °C, (**b_2_**) 1400 °C, (**c_2_**) 1450 °C, (**d_2_**) 1500 °C, (**e_2_**) 1550 °C.

**Figure 7 materials-15-03471-f007:**
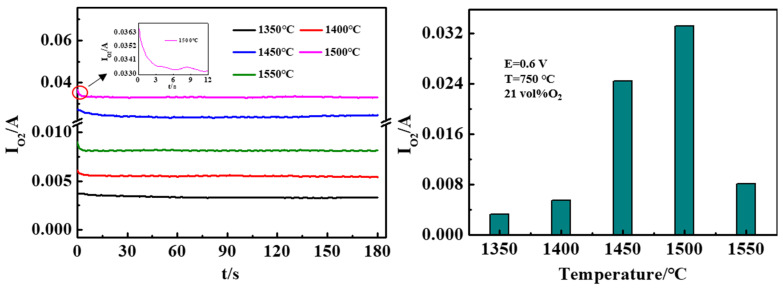
Chronoamperometric curves of Pt/YSZ electrodes at different sintering temperatures (**left**) current versus time; (**right**) steady current value.

**Figure 8 materials-15-03471-f008:**
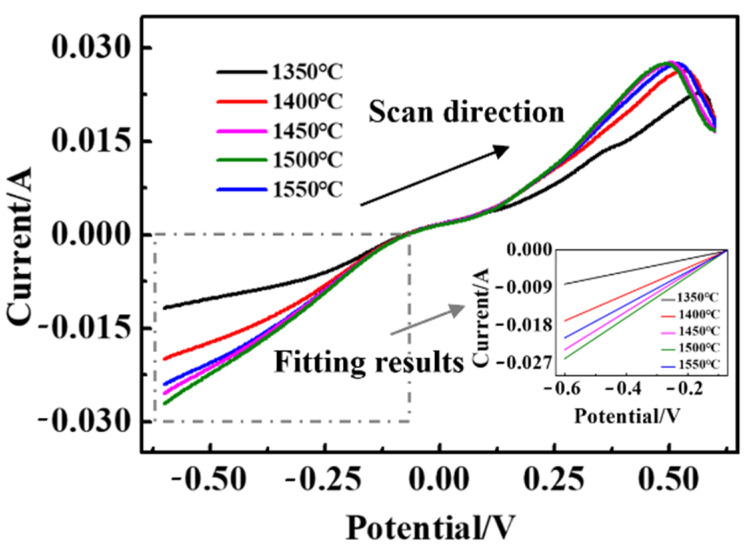
LSV curves of the Pt/YSZ electrodes.

**Figure 9 materials-15-03471-f009:**
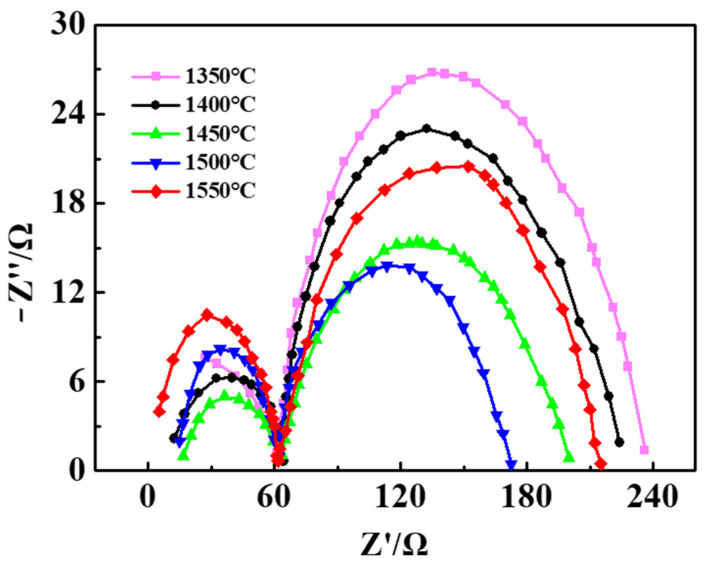
AC impedance spectrums of the Pt/YSZ electrodes sintered at different temperatures tested at 750 °C.

**Figure 10 materials-15-03471-f010:**
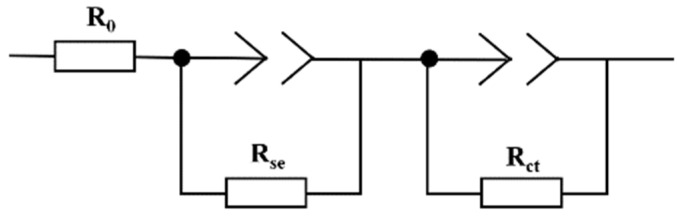
Equivalent circuit diagram of AC impedance.

**Figure 11 materials-15-03471-f011:**
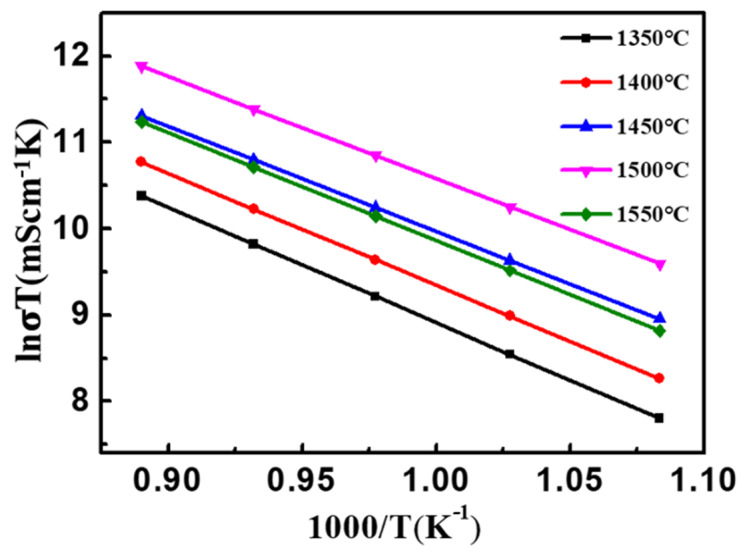
Arrhenius curves of Pt/YSZ electrodes at different sintering temperatures.

**Table 1 materials-15-03471-t001:** Physical properties of YSZ and Pt materials.

Materials	Grain Size (nm)	D_50_ (µm)	Specific Surface Area (m^2^/g)	Phase Analysis
YSZ(Zr_0.95_Y_0.05_O_1.975_)	42.6	0.169	12.83	c-ZrO_2_
Pt	79	0.5	13.56	c-Pt

**Table 2 materials-15-03471-t002:** Activation energy of electrode reaction at different sintering temperatures.

sintering temperature/°C	1350	1400	1450	1500	1550
electrode activation energy/eV	1.16	1.12	1.05	1.02	1.08

## Data Availability

All data are available from the corresponding author on reasonable request.

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
