# Peer review of "Effect of Sintering Temperature on Adhesion Property and Electrochemical Activity of Pt/YSZ Electrode"

_materials, 2022, doi:10.3390/ma15103471_

Round 1
Reviewer 1 Report
This work is devoted to studying the effect of sintering temperature up to 1550°C on the characteristics of the Pt/YSZ electrode to improve NOx sensors parameters. The development of highly sensitive, stable and commercially effective nitric oxide sensors remains an actual problem. There are many papers currently published on this topic. In this paper, the authors, based on the existing technology for the production of conductometric type nitrogen sensors, explore possible ways to improve the operating parameters of the sensors by changing the sintering temperature.
The research topic is relevant to the Materials journal scope.
It is suggested to solve the following problems before publishing:
- Figure 1: The diagram is not very clear, it seems that a tube for gas inlet is indicated, but it is not clear how everything is connected. It would also be useful to increase the font of the captions, not easy for reading now;
- It is not clear why authors interpret the composition of the phases from the SEM data on Figure 1. It is necessary either evidence-based explanations in the text or confirming experimental data (for example, energy dispersive X-ray spectroscopy).
- The results presented in Section 3.2 - SEM morphology of the interface between Pt/YSZ electrodes and YSZ electrolyte at different sintering temperatures are, in my opinion, interesting and of great practical importance. But is it correct to call such studies - studies of adhesion, because the paragraph contains only the results of morphology, the results of the study of adhesion are not given. Thoughts have been made about the optimal structure, but that they will actually demonstrates better adhesion has not been proven. Here it is necessary to present the results somehow more accurately, perhaps without pretending to studies of adhesion.
- many typos and indexes of degrees that are "out of place".
Reviewer 2 Report
In this manuscript, the effect of sintering temperature on the adhesion of Pt to YSZ support was presented, what could be employed for an electrochemical NOx detection in the exhaust gases through measuring of the oxygen anion oxidation current.
The results are not especially revealing, but more importantly the interpretation of the electrochemical experiments results is ill-conceived.
The manuscript is no suitable for publication.
Specific comments:
line 157
Oatoms reached the Pt/YSZ electrode and react with Ptatoms to form Pt-O. The conductivity of Pt-O is very poor, which hinders the
charge transfer process and reduces the current density ...
comment:
The claim of the Pt-O formation is not supported by any evidence. Moreover, if the non-conductive Pt-O were to get formed, then there would be a continous decline of the current recorded in the CA experiments (Fig5). Evidently, Authors assume that only a small portion of Pt grains gets passivated (Pt-O), while the rest of their surface is still available for oxygen evolution (reaction 2).
As a matter of fact one can discern a current decline, lasting a couple of seconds, just after the potential was switched on. In my opinion this decline was caused by oxygen anion transportation lag (diffusion and migration limitation). The initial current decrease is almost absent for the samples prepared at low temperatures, i.e. 1350 and 1400 °C, what must be somehow related to the structure differences between them, e.g. porosity/specific surface, induced by the sintering temperature.
I would also suggest considering a following mechanism:
O^2- + Pt -> Pt-O + 2e - electrochemical step
2Pt-O --> O2 + 2Pt - chemical step
line 177
The speeds
comment:
The rate
line 187
However, no cathodic peak is observed. It could be explained that the cathodic reaction did not rely on the reduction of Pt-O.
and
line 246
O2- in YSZ is converted into chemisorbed O2 on Pt/YSZ, which is desorbed into the gas phase in the form of molecular oxygen...
comment:
How could you expect oxygen to take part in the reduction process when it is gone?
Reviewer 3 Report
Despite interesting and well written, there are many reports on this topic, such as:
Jaccoud, A., Fóti, G., Wüthrich, R., Jotterand, H. and Comninellis, C., 2007. Effect of microstructure on the electrochemical behavior of Pt/YSZ electrodes. Topics in Catalysis, 44(3), pp.409-417.
Sekhar, P.K., Brosha, E.L., Mukundan, R., Nelson, M.A., Toracco, D. and Garzon, F.H., 2010. Effect of yttria-stabilized zirconia sintering temperature on mixed potential sensor performance. Solid State Ionics, 181(19-20), pp.947-953.
Buyukaksoy, A., Petrovsky, V. and Dogan, F., 2013. Solid oxide fuel cells with symmetrical Pt-YSZ electrodes prepared by YSZ infiltration. Journal of the Electrochemical Society, 160(4), p.F482.
Just to cite a few. The innovation with respect to previous work should be made clearer. That must be done both in the discussion and in the conclusions.
In "The YSZ green tapes were prepared by tape casting process. Triethanolamine (AR) 79 ,polyvinyl butyral (PVB, AR)", there seems to be an error with an extra space before the comma in line 79 (page 2).
Writing "3. Test Results" is not usual.
The visual arrangement of Fig. 4 is a bit confusing.
In Equation 4, where did the Authors take sigma from? It should be better explained, since the electrical model is much more complex than a resistor.
The way the conclusions are written are more alike a presentation than a paper.
Finally, there are many many phrases that are too long (search for phrases with multiple lines, such as > 4). In most cases, it makes them harder to read.
Round 2
Reviewer 2 Report
Dear Authors,
Response 3
Oxygen undergoes a reduction process of oxygen atoms on one side of the (Pt/YSZ electrode)/YSZ electrolyte sensor unit, and oxygen anions are transported through the oxygen vacancies of the YSZ ...
comment:
The problem we are discussing about has nothing to do with the principle of pumping oxygen, it deals with your CV experiments, which did not show the expected platinum oxide reduction. You write:
'The cathodic reaction is the reduction process of Oatoms, the cathodic reaction model can generally be expressed as [24]:
PtO + 2e --> Pt + O2-'
The second qutoted by you paper [25] also reported appearance of the cathodic peak: 'According to literature [15–17,19–21], we assign the cathodic peak to reduction of Pt oxide at the interface Pt/YSZ, which was formed at VWR = 0.5 V.'
So, what was the problem with your CV experiment? You did recorded the reduction current but you can not assign any reaction to it for the system composed of only 2 electroactive species: PtO, O2.
Maybe, you should have polarized the electrode further towards much more negative potentials?
I would also like to raise another question.
Revised version. Line 185
As a matter of fact this fragment remained almost unmodified compared to the first version of the manuscript.
The conductivity of Pt-O is very poor, which hinders the charge transfer process and thus reduces the current density [25]. However, the number of platinum atoms on the electrochemical reaction site is limited, and the reaction reaches saturation in a short time [24]. At this time, the anodic polarization proceeds through the reaction (2), that is, the reaction moves to the interface between the Pt/YSZ electrode and the YSZ electrolyte,
comment:
This passage is not clear enough to me. If platinum gets passivated, and becomes non-conductive, then how do electrons from oxygen anions reach the external electric circuit?
Reviewer 3 Report
The previously suggested corrections have been kindly made by the Authors and the paper can now be published.
Author Response
Thanks again for your selfless work. It truly is a privilege that your valuable comments made the manuscript more rigorous.
Round 3
Reviewer 2 Report
I am satisfied with the clarifications made by authors and recommend current version of the manuscript for publication.